# A Comprehensive Cheminformatics Analysis of Structural Features Affecting the Binding Activity of Fullerene Derivatives

**DOI:** 10.3390/nano10010090

**Published:** 2020-01-02

**Authors:** Natalja Fjodorova, Marjana Novič, Katja Venko, Bakhtiyor Rasulev

**Affiliations:** 1National Institute of Chemistry, SI-1000 Ljubljana, Slovenia; marjana.novic@ki.si (M.N.); katja.venko@ki.si (K.V.); 2Department of Coatings and Polymeric Materials, North Dakota State University, Fargo, ND 58102, USA; bakhtiyor.rasulev@ndsu.edu

**Keywords:** fullerene derivatives, drug-like descriptors, binding activity, cheminformatics, neural networks modelling, hydrogenation, pharmacology, toxicology

## Abstract

Nanostructures like fullerene derivatives (FDs) belong to a new family of nano-sized organic compounds. Fullerenes have found a widespread application in material science, pharmaceutical, biomedical, and medical fields. This fact caused the importance of the study of pharmacological as well as toxicological properties of this relatively new family of chemicals. In this work, a large set of 169 FDs and their binding activity to 1117 disease-related proteins was investigated. The structure-based descriptors widely used in drug design (so-called drug-like descriptors) were applied to understand cheminformatics characteristics related to the binding activity of fullerene nanostructures. Investigation of applied descriptors demonstrated that polarizability, topological diameter, and rotatable bonds play the most significant role in the binding activity of FDs. Various cheminformatics methods, including the counter propagation artificial neural network (CPANN) and Kohonen network as visualization tool, were applied. The results of this study can be applied to compose the priority list for testing in risk assessment related to the toxicological properties of FDs. The pharmacologist can filter the data from the heat map to view all possible side effects for selected FDs.

## 1. Introduction

Fullerene derivatives (FDs) are a relatively new class of organic compounds belonging to nano-sized materials. These materials have opened up new opportunities in nanotechnology, as well as in medicine [1].

Fullerenes are commonly classified as “radical sponges” [2] due to their remarkable reactivity with free radicals [3,4,5,6]. The radical scavenging properties of fullerenes have found many applications in biological systems. They were applied for treatment of various free radical-induced biological disorders, including mostly neurodegenerative diseases (i.e., amyotrophic lateral sclerosis, Alzheimer’s disease, Parkinson’s disease) and other cytotoxic processes caused by oxidative stress. The fullerenes have found the application as cytoprotective agents against oxidative stress [7]. The FDs can prevent apoptosis by neutralization of reactive oxygen species (ROS). The antimicrobial property of FDs was demonstrated in [8] where authors showed the inhibition of *Escherichia coli* bacteria growth by FDs. The ability of fullerenes to fit inside the hydrophobic cavity of HIV proteases makes them a potentially good inhibitor of the catalytic active site of enzyme. Therefore, FDs have found their application as antiviral drugs [9,10,11,12,13,14]. The antiviral activity of FDs was found to be due to the antioxidant activity of them. At the same time, when fullerenes are exposed to a light, they can initiate formation of ROSs (singlet oxygen and superoxide), which leads to antibacterial/antimicrobial activity, and this effect of FDs is used in water treatment systems [11,15,16,17,18,19].

FD nanostructures can be used in many applications. The details about synthesis, chemistry, and application of fullerenes were reported in several reviews [6,20,21]. Toxicological studies of fullerenes were reported in [22]. Thus, pristine fullerenes have shown a low toxicity. At the same time, there is still a lack of knowledge related to toxicity of FDs per se.

Nanoparticles, including fullerenes, often pose a serious threat to human health, the environment, or both. Nanoparticles can cause toxic effects at different levels: Cellular, subcellular, and bio molecular [23,24]. In this regard, FDs also can have a significant impact on environment and human health and; therefore, these nanostructures need to be investigated as well for potential toxicological and environmental risk.

There is still a lack of knowledge about toxicity of FD nanostructures and their mechanisms of action in living organisms. To tackle this problem the research related to activity/safety of this class of chemicals is initiated in this work. 

The novel approaches for risk assessment of nanomaterials using computational tools, like quantitative structure activity relationships (QSARs), are discussed in several publications [25,26,27,28,29,30,31,32]. Thus, reliable QSAR models can offer a time-effective and cost-effective measure of chemicals’ properties in the absence of new experimental data. As per FDs, there are a number of computational studies and the application of cheminformatics tools including QSAR models for modelling and prediction of FDs’ properties, including HIV protease inhibition, which is also discussed in articles [33,34,35].

In last years, the risk assessment of chemicals has focused on the mechanistic interpretation of QSAR models based on description of the relationship between the descriptors used in a model and the investigated endpoint. This task can be also solved using recently developed drug-like descriptors [36]. The concept of drug-like properties is a hot topic currently [36]. Drug-like descriptors brought to light the understanding of the behavior of chemicals in living organism in the terms of absorption, distribution, metabolism, and excretion (ADME) processes, which are related to pharmacokinetic and/or pharmacodynamics processes [37,38]. Therefore, in the current study, we applied the drug-like descriptors related to FDs and considered the correlation between these descriptors and binding activity. Moreover, the understanding of the relationship between the chemical descriptors (which express electronic, topological, geometrical, and other properties) and substituents (functional groups) of FDs was the focus of the current investigation.

In the article by Andrew Worth «The future of *in silico* chemical safety … and beyond», it was pointed out that the new term that has gained acceptance is *in chemico*, referring to abiotic assays that measure chemical reactivity (toward proteins) [39]. Therefore, in the current article, the binding activity of studied FDs with functional groups was explored in terms of possible toxic and pharmacological impact. We suppose that investigation of the binding activity of FDs can bring a new knowledge to the global activity of this class of chemicals covering a possible activity with 1117 disease-related proteins. The present study paves a way to select the most significant interactions between FDs and proteins and then to perform an individual testing of the best FDs. The article was also focused on the main groups of proteins and their role in living organisms.

A comprehensive interpretation of a large dataset of 169 FDs requires an understanding about the content of the dataset related to a type of functional groups connected to fullerene core. In the current study, the FDs were classified depending on the type of bonding to fullerene core. Then the analysis was made on the most active and the least active FDs based on the presence of functional groups. The investigated activity was expressed as binding score that demonstrates the ability of FDs to interact with certain proteins. The performed study demonstrated how structural changes of FDs lead to changes in investigated property (i.e., binding activity). A wide variety of 169 FDs’ structures by different types and classes gave valuable information about the properties of FDs related to their potential application as drugs, simultaneously with evaluation of their potential safety/toxicity. To visualize the drug-related properties and safety/toxicity predictions, the techniques based on CPANN algorithm and Kohonen networks were applied. A heatmap of binding scores for 169 FDs with 1117 disease-related proteins represents valuable information on the individual selection of possible drugs, together with information about the possible side effects and toxic properties of considered individual FDs. It should be noted that not only functional groups were the focus of our study, but also the level of hydrogenation (saturation) of FDs and its influence on binding activity.

Moreover, in this study, analysis of structural descriptors (drug-like descriptors) and their connection with structure and binding activity of FDs was done. Application of cheminformatics tools enabled explanation of some features of FDs affecting the binding activity. 

## 2. Materials and Methods 

### 2.1. Dataset 

A large set of 169 fullerene derivatives (FDs), represented in our previous work [40], was explored in the current study. The list of 169 FDs is represented in the Appendix A. The classification of FDs dependent on the type of bond related to substituent group attached to the fullerene C60 core is represented in Appendix A. The investigated FDs include fullerenes C60, C70, and C80.

In a previous work [40], the binding activity for 169 FDs related to 1117 proteins, expressed as a binding score (Bscores), were calculated. The proteins were extracted from the RCSB Protein Data Bank [41]. A heat map of the binding scores activity of 169 FDs with 1117 proteins is represented in Appendix A. In this file the Bscores of 1117 proteins data were screened against 169 FDs, forming the color coding heatmap, where red-orange color corresponds to the highest value of Bscores, yellow—the moderate values, and green—the lowest values of Bscores. 

Let us consider the overall characteristics of these 1117 disease-related proteins. It is well known that proteins perform many different functions within organisms, such as (a) catalyzing metabolic reactions; (b) DNA replication; (c) responding to stimuli, providing structure to cells and organisms; and (d) transporting molecules from one location to another. The proteins investigated in this study belong to different classes according to their functions. The short description of groups of proteins related to different functions in organism, in a broad sense, is represented in Table 1.

The detailed description of the individual groups of proteins, listed in Table 1, is provided in the Appendix A.

### 2.2. Descriptors

Twenty five descriptors generated with use of DataWarrior software (developed by Actelion Pharmaceuticals Ltd. (Allschwil, Switzerland) [42]) were employed in the study: H-acceptors, H-donors, total surface area, relative PSA, polar surface area, drug-likeness, Mol_weight, cLogP, cLogS, electronegative atoms, stereo centers, rotatable bonds, rings closures, small rings, aromatic rings, aromatic atoms, sp3-atoms, symmetric atoms, amides, amines, aromatic nitrogen, basic nitrogen, acidic oxygens, non-H atoms, non-C/H atoms. These descriptors are used in drug design to describe drug-like properties of chemicals. Additionally, two descriptors employed by authors [40]—polarizability in cubic angstroms (*QPpolrz*) and topological diameter (TD) characterized the size of molecules—were used in the study.

### 2.3. The Counter Propagation Artificial Neural Network (CPANN) Algorithm and Self-Organizing Kohonen Maps

The detailed description of CPANN employed in the current study is represented in the Appendix A. The architecture of CPANN is shown in Appendix A and reported in papers [43,44,45].

The CPANN is basically a two-layer neural network. It consists of a Kohonen layer (influenced by the input (descriptors)) and an output layer (influenced by the target (binding activity—Bscores)). CPANNs can be used as lookup tables. There is a one-to-one correspondence between the neurons in the Kohonen map and those in the output map. The self-organizing Kohonen Maps, as a data visualization technique [46], was applied for visualization of structurally similar molecules that tend to have similar activities. In this work, the focus was on the relationship between descriptors and FDs binding activity. Moreover, the clusters of FDs with similar functional groups and structure were examined in relation to the binding activity. The description of the code used in this study and its applications are reported by Grošelj et al. [47].

### 2.4. A Self-Organizing Kohonen Network

Kohonen networks learn to create maps of the input space in a self-organizing way. The best-known and most popular model of self-organizing networks is the topology-preserving map proposed by Teuvo Kohonen [46]. Self-organizing maps (SOMs) belong to a group of neural networks that use unsupervised learning [46]. Unsupervised learning supports the arrangement of objects (proteins in the present work) in the input layer map based on the similarity among input variables (expressed as binding scores activity). There is no golden standard and no straightforward external validity testing in this case. Each SOM consists of a predefined number of neurons, where each neuron has an associated weight vector. The number of elements in each vector is equal to the dimensionality of the input space. In our study, we used 5 × 5 matrices. The similar objects placed in the same or closest neurons. In the study, the Kohonen network was applied for organization of 1117 proteins in 2D map, and the number of proteins was reduced by elimination of the most similar objects. 

## 3. Results

### 3.1. Reduction of the Number of Proteins by Using Kohonen Network

In this part of study the matrix containing the binding activity, expressed as a binding scores (Bscores), for 1117 proteins related to 169 FDs was composed. The neural network with dimension 5 × 5 was trained for 100 learning epochs. A Kohonen map of 5 × 5 with the distribution of 1117 proteins was obtained. The objects (protein numbers) were distributed by similarity determined by the binding activity. For example, similar objects are located close to each other in the Kohonen map. The Euclidean distances for each of the proteins placed in individual neurons were calculated. The Figures with Euclidean distance vs. ID of proteins were placed in the Appendix A. In each neuron, the proteins with the smallest and largest Euclidean distances were selected. This operation was done to reduce the number of proteins and get a global generalized view. This helped to find the main classes of proteins which differ by their behavior related to the considered FDs. The number of proteins were reduced from 1117 to 57. The Kohonen map with distribution of 57 proteins is represented in Figure 1.

The substances like FDs can interact with proteins in a different way, depending on the type of proteins and their function, and can cause the appropriate changes in system. The interaction of proteins with FDs can lead to (a) change in catalyzing the metabolic reactions in organism; (b) chemical changes to the genetic material; (c) change in responding to stimuli, providing structure to cells and organisms; or (d) change in transporting molecules from one location to another. 

The obtained Kohonen map in Figure 1 contains enzymes (red color), receptors (violet color), gene regulation or transcription (green color), and transport proteins (blue color).

### 3.2. Selection of Characteristic for Binding Activity

The comprehensive multi-software protein-ligand docking simulation and chemoinformatics approaches were applied to investigate the interaction of 1117 proteins with 169 FD nanoparticles, which was reported first in the work [40]. The binding score (Bscores) parameter is responsible for several types of intermolecular interactions and apprises the force of interaction between protein and ligand (FD). Firstly, the average value of Bscores for 1117 proteins (marked as Average sum) calculated for each of 169 FDs was found. Secondly, a top 110 proteins that possessed the highest binding activity was selected and determined the average Bscores for these 110 proteins (marked as Average110). Thirdly, the average Bscores for 57 proteins was determined through selection from the Kohonen map (marked as Average 57). These 57 proteins cover the main functions in the existing dataset (see Figure 1). 

The structures of FDs were expressed as two descriptors: Polarizability in cubic angstroms (*QPpolrz*) and topological diameter (*TD*) selected in our previous study [40]. 

Then, the correlation between Average sum (for 1117 proteins), Average 110, Average 57, and *QPpolrz* and *TD* was the focus of the investigation. The results are shown in Table 2. 

Obtained results demonstrated that average Bscores for the 1117 proteins, the top 110 proteins, and for 57 proteins are highly correlated with the structure of FDs expressed as two descriptors: *QPpolrz* and *TD*.

In the next step, the Average sum for all 1117 proteins was used as a characterization of binding activity of FDs and marked as “Av Bscores”. 

### 3.3. The Characteristics of FDs Dataset with Relation to Binding Activity 

The current list of FDs covers many classes of chemical compounds. In this section the binding activity was expressed as average binding scores (Av Bscores) and considered here in the range from 4000 to 8000. The distribution of 169 FDs dependent on their binding activity (Av Bscores) is represented in Figure 2.

The FDs with binding scores in the range 5500 to 8000 are the most active (sector 1). The interval 5500 to 5000 belongs to the moderately active FDs (sector 2). The majority of FDs are located in this sector. The least active or low active FDs have the binding activity in the scale of 5000 to 4000 (sector 3).

The goal of this work was to illustrate how functional groups of FDs influence on the binding activity (Av Bscore) of FDs. The FDs were considered in the order of decreasing the Bscores values.

The classification of FDs depending on the type of bond of substituent groups attached to the fullerene C60 core with indication of the level of binding activity is represented in the Appendix A. The considered dataset of FDs contains the FDs substituent groups attached to C60 core with (1) single bond containing alkyl groups; (2) cyclopropane three-membered ring; (3) pyrrolidine (five-membered ring) or pyridine (six-membered) heterocycles containing nitrogen; (4) six-membered (cyclohexane) ring); (5) benzene (aromatic six-membered) ring; (6) fused pair of six-membered rings; and (7) bridged bicycle rings which are illustrated in Appendix A.

### 3.4. The Characteristics of the Least Active and the Most Active FDs

The lowest value of binding activity belongs to pristine fullerene C60. The surface of C60 fullerene contains 20 hexagons and 12 pentagons, where all rings are fused, all double bonds are conjugated. In spite of their extreme conjugation, they behave chemically and physically as electron-deficient alkenes rather than electron rich aromatic systems [48]. The least active FDs are shown in Table 3.

The FDs without functional groups or with only few hydroxyl –OH groups or carboxyl groups –COOH demonstrated the lowest binding activity (Bscores), approximately less than 5000.

Table 3 also shows the differences between the binding activity (Bscore) of pristine fullerene C_60_ and the least active fullerenes.

The least active is fullerene C_60_ (**FD168** = 3938.3). The differences in binding activity between C_60_ (**FD68** = 3938.3) and C_70_ (**FD50** = 4224.3) was equal to 286, and between C_60_ (**FD168** = 3938.3) and C_80_H_2_ (**FD169** = 4398.5) was equal to 460.2. The addition of four hydroxyl groups (–OH) in **FD160** (4192.2) resulted the increase of binding activity by 253.9 in comparison with C_60_ (**FD168** = 3938.3). The binding activity of saturated **FD61** (4725.1) containing 20 hydroxyl groups is greater by 786.8 in comparison with C_60_ (**FD168** = 3938.3). Approximately the same rise of binding activity, equal to 797, in comparison with C_60_ (**FD168** = 3938.3) was obtained for **FD93** (4735.3) containing 24 hydroxyl groups. The activity of **FD93** (4735.3) containing 24 –OH, in comparison with **FD61** (4725.1) containing 20 –OH, was increased only by 10. We can suggest that in this case the presence of –OH has influence on activity, but the level of saturation of considered FDs should be taken into consideration too. Thus, the higher level of saturation of **FD61** (C60H34(OH)20) containing 34 hydrogens in C60 core) in comparison with **FD93** (C60H26(OH)24), containing 26 hydrogens in C60 core) can cause the reduction of binding activity of **FD93**. Increasing the binding activity of **FD93** caused by the additional four hydroxyl groups –OH can be overlapped with decreasing activity due to a lower level of saturation (26-H in **FD93** in comparison with 34-H in **FD61**). 

Allen and co-workers conducted investigations in order to determine the antioxidant activity of a range of fullerenes (i.e., C60, C70, and fullerene soot) and then ranked them according to their comparative efficiency. They proposed the following sequence of antioxidant efficiency: C70 > C 60/C70 (80/20) > C60/C70 (93/7) > C60 [49]. The binding activity of C70 in comparison with C60 was higher by 286. In addition, we supposed that theoretically calculated Bscores can also be an indicator of antioxidant activity of FDs. 

The differences in binding activity (∆Bscores) between the most active FDs and pristine fullerene C_60_ are represented in the Figure 1. The presence of the most active groups in FDs lead to an increase of binding activity in comparison with pristine fullerene C_60_ by 2162–3947. Thus, the differences in binding activity (∆Bscores) of the moderate to least active FDs in comparison with pristine fullerene C_60_, approximately, is in the range 2162–475.

Analyzing data represented in the Appendix A, one can notice that the most active **FD6** has the longest alkyl chain with the alkenyl group (double unsaturated C = C-) in the middle of the chain (see Appendix A). Appendix A, containing two benzene rings, had high activity. In Appendix A the activity was caused by presence of ammonium groups (**FD**s **163**, **162**, **161**), and in Appendix A by presence of phosphonate groups **FDs 158**, **165**. Aromatic nitrogen in Appendix A caused the high activity for **FDs 154**–**157** and **112**, **113**. Pyridine rings as well as benzene rings, probably, contribute to the high binding activity in Appendix A (see **FDs 35** and **36**).

**FDs 118**–**124** in Appendix A possessed high activity due to presence of 8–14 NO_2_ groups. FDs from the Appendix A experienced high activity due to presence of pyridine groups (**FDs 35**, **36**). 

The analysis of the activity of clusters of considered FDs is given below using visualization tools in 2D Kohonen maps, which is complemented with analysis of structural drug-like descriptors correlated with the binding activity.

### 3.5. CPANN Model Based on Drug-Like Descriptors

#### 3.5.1. Selection of Drug-Like Properties Descriptors for Modelling

Drug-likeness is a qualitative concept used in drug design related to bioavailability of substances. It is estimated from the molecular structure. Drug-likeness may be defined as a complex balance of various molecular properties and structure features which determine whether a particular molecule is similar to the known drugs. We notice the absence of studies about drug-likeness of FDs. DataWarrior software developed by Actelion Pharmaceuticals Ltd. [43] has been used to calculate drug-like descriptors for FDs. Analysis of drug-like properties of FDs can be related to the novelty of our study. 

Between properties considered for drug-likeness, we can highlight that the hydrophobicity, electronic distribution, hydrogen bonding characteristics, molecule size and flexibility, and of course, the presence of various pharmacophore features influence the behavior of molecule in a living organism, including bioavailability, transport properties, affinity to proteins, reactivity, toxicity, metabolic stability, and many others. The concept of drug-likeness provides useful guidelines for early-stage drug discovery [37,50]. It contains the analysis of the observed distribution of some key physicochemical properties of approved drugs, including molecular weight, hydrophobicity, and polarity related to known drugs [38]. These parameters are well known and applicable in drug design. This is why it is sensible to use these criteria in the study of a new class of chemicals, like FDs, to find some features understandable for drug design researchers and for future examination of a unique class of chemicals that are promising for application in drug design. The assessment of drug-likeness is known as Lipinski’s rule of five (Ro5), where simple count criteria (like limits for molecular weight, log P, or number of hydrogen bond donors or acceptors) and others were used [51]. 

Twenty-five drug-like descriptors were calculated using DataWarrior software [42]. These descriptors are reported in the section “Materials and Methods”. The “drug-like” properties include the structural features and physicochemical properties. These properties can be used for characterization of pharmacophore: A substituent in FDs or a part of a molecular structure that is responsible for particular biological or pharmacological interaction [52].

#### 3.5.2. CP ANN Model Based on 27 Descriptors

The CPANN model developed in this section was based on twenty-seven molecular descriptors. First, twenty-five descriptors were generated by DataWarrior software, where all calculated descriptors are drug-like properties of compounds. These descriptors can be used to explore some properties that might play a role in formation of the interaction between the ligand (FD) and receptor (protein). Two descriptors *QPpolrz* and *TD* that were applied in the previous study [40] were added in the current study.

The CPANN models based on the generated drug-like descriptors were trained. The input data for 169 FDs were normalized. The optimal model was obtained with dimension 20 × 20 and number of learning epochs equal to 100. The model demonstrated the following statistical performance for the whole data set that was used as a training set: squared regression coefficient, R^2^ = 0.96, (RMSE = 0.21), leave-one-out cross-validation (LOO-CV) regression coefficient, Q^2^_cv_ = 0.87, (RMSE = 0.22). The internal validation of CPANN models was performed using the LOO-CV procedure for evaluation of the quality and goodness of fit of the model [53,54].

It should be highlighted that the developed model was used for visualization of the whole dataset of FDs and to study the correlation and relationships between descriptors and binding activity.

#### 3.5.3. Analysis of Distribution of FDs in the Top Map of the CPANN Model Based on 27 Descriptors

The distribution of FDs in the top map 20 × 20 of the CPANN model based on 27 descriptors overlapped with the output layer, with distribution of values of binding activity, is represented in Figure 3.

The blue color in Figure 3 corresponds to the lowest value of binding activity (Bscores) and the red color to the highest values. The scale bar represents the normalized data of Bscores. The top map in Figure 3 is complemented with illustration of some of the most active groups of FDs and weight maps of related descriptors that have extreme values for these selected groups of FDs. The most active group of FDs located in neurons 1 × (7,8,10) contain ammonium NH_3_^+^ groups. This area corresponds to the highest value of descriptors: basic nitrogen and rotatable bonds. See the weight maps of basic nitrogen and rotatable bonds at the left bottom corner in Figure 3. 

The active group of FDs attached to the C60 core with cyclopropane three-membered ring and containing two benzene rings is located in neurons 4 × 12 and 5 × (11,12). This group of FDs corresponds to the area with the highest values of topological diameter descriptor. The weight map of the topological diameter descriptor is shown at the left top corner in Figure 3. 

The group of FDs connected to C60 core with benzene ring and containing from 14 to 8 NO_2_ groups belongs to the most active FDs too, and is located in the right top corner of top map in neurons 20 × (20,19,17) and 19 × 19. These groups of FDs are characterized by the highest values of descriptors: electronegative atoms and acidic oxygen. The weight maps of pointed descriptors are shown at the right top side in Figure 3. 

The next active group of FDs contains the FDs connected to C60 core with pyrrolidine ring and containing nitro aromatic substituents. This group of FDs is shown in Figure 3 at the right bottom corner of the top map. The highest values of the topological diameter descriptor correspond to this area, which was highlighted in the weigh map located close to this group from the right bottom side of Figure 3. 

Of course, Figure 3 only partly demonstrates the characterization of some groups of FDs and the related descriptors that play the most significant role only for these groups. 

To clarify this statement, let us consider the most active FDs with the highest value of Bscores. These are **FD162** and **FD163**, containing 6 and 8 ammonium NH_3_^+^ groups, correspondingly, and located in neurons 1 × (7,8). These FDs possess the highest value of molecular weight, rotatable bonds, sp3 atoms, amines, basic nitrogen, non-H atoms, and polarizability *QPpolrz*. 

The lowest value of binding activity (Bscores) corresponds to pristine fullerene **FD168** (C60) located on neuron 8 × 20. The fullerene C60 possesses the lowest total surface area, molecular weight, rotatable bonds, electronegative atoms, sp3 atoms, polarizability *QPpolrz*, and topological diameter. Thus, C60 does not have functional groups or substituents and can be used a reference molecule for comparison with other FDs containing functional groups related to binding activity.

The present section demonstrated only a quick simplified look on information about the relationship between FDs, their property (BScores), and applied descriptors supported by visualization tools (particularly CPANN).

#### 3.5.4. Consensus Model Based on 10 Descriptors 

The key task of QSAR models is the selection of a minimal set of descriptors correlated with the studied property of an organic compound [55]. Reduction of the number of descriptors was performed using the Kohonen mapping technique. This method is based on selection of descriptors from the same neuron with the greatest and smallest Euclidian distances, as the inherent property of the Kohonen map is the position of the same or similar objects at the same or closest neurons. 

Selection of descriptors was made using a 2 × 2 Kohonen map. Eight (8) descriptors were selected: cLogS, sp3-atoms, drug-likeness, aromatic atoms, electronegative atoms, rotatable bonds, amides, basic nitrogen. 

The consensus model was built using eight drug-like properties descriptors plus two descriptors from previous study of polarizability in cubic angstroms and topological diameter (*QPpolrz* and TD) [40]. The input data were normalized. Binding activity (Bscores) were used as a response. The optimal selected model with architecture 20 × 20 was trained for 100 learning epochs. 

The model demonstrated a high level of accuracy with R^2^ = 0.95 and leave-one-out test cross-validation 0.92. 

The output layers for responses were used in the study as weight maps, for demonstration of distribution of these parameters in 2D space, for comparison with distribution of applied descriptors. This technique represents the distribution of multidimensional parameters in 2D space and allows for analysis of the similarities or correlation between studied parameters.

The consensus CPANN model mentioned above was used for visualization of molecular space in 2D maps, and analysis of the distribution of FD molecules decorated with different functional groups related to different binding activity. The top map 20 × 20 of the CPANN model based on 10 descriptors with distribution of FDs overlapped with the output layer of binding activity (Bscores) is represented in Figure 4.

#### 3.5.5. Analysis of Distribution of FDs in the Top Map of the Consensus Model Based on 10 Descriptors.

The top map was complemented with illustration of some of the most active groups of FDs as well as the FDs with lowest activity. 

The color in the map (Figure 4) depends on the binding activity. The red one was related to the most active space (the highest value of BScores), while the dark blue area corresponded to the least active FDs, having the lowest value of Bscores.

The classification of FDs dependent on type of bond of substituent groups attached to the fullerene C60 core with detailed characterization of activity (A-active, M-moderate active, and L-low active) is represented in the Appendix A. 

The most active groups of FDs shown in the Figure 4 are described below. Group 2 is located in Appendix A. For reference see group 2a (Group 2a attached to the C60 core with cyclopropane three-membered ring and containing two benzene rings with level of activity 7257–6164) (Appendix A); group 2c (Group 2c attached to the C60 core with cyclopropane three-membered ring and containing ammonium groups with level of activity 7885–6766), and 2d (Group 2d attached to the C60 core with cyclopropane three-membered ring and containing phosphonate groups with level of activity 6621–5975) (Appendix A). 

Group 3 is located in Appendix A. For reference see group 3a (Group 3a attached to the C60 core with pyrrolidine (five-membered ring) and containing aromatic nitrogen with level of activity 6802–6230) (Appendix A); group 3c (Group 3c attached to the C60 core with six-membered cycle and containing in the side chain pyridine ring or attached to the C60 core with pyridine ring or six-membered heterocycle containing nitrogen with level of activity 6709–5650) (Appendix A). 

Group 5 is located in Appendix A. Group 5b1 (Group 5b1—the most active FDs connected to C60 core with benzene ring and containing 8–14 nitro groups -NO2 with activity 7120–5886) (Appendix A).

The low active FDs are located in the center of the Kohonen map (Blue area) and are listed in the top right corner, with indication of the level of their activity expressed as Bscores.

### 3.6. The Characterization of Drug-Like Descriptors Related to Binding Activity

#### 3.6.1. Results of Principle Component Analysis for 25 Drug-Like descriptors 

The principle component analysis (PCA) was performed for 25 drug-like descriptors. The role of descriptors is illustrated at the loading plot for the first two components where the loadings for the second component (*y*-axis) are plotted versus the loadings for the first component (*x*-axis). A line is drawn from each loading to the (0, 0) point. The loading plot represented in Figure 5 represents a good visualization for 25 drug-like descriptors and their relation to each other. The highly correlated descriptors with CC from 1 to 0.895 are located on the right side of the plot, labeled as red lines. The aromatic atoms and aromatic rings, which have the negative correlation with sp3-atoms and stereo centers are located in opposite side from 0.0 in Figure 5. The cLogP is located opposite to total surface area. 

In the study below the correlation between these descriptors was performed using weight maps of CPANN models and by calculating correlation coefficient using the Minitab program.

#### 3.6.2. How Drug-Like Descriptors Correlated to Binding Activity 

The correlation between some of drug-like descriptors is represented in Figure 6. First of all, the Pearson correlation coefficients (CC) were calculated for all applied descriptors as well as for binding activity (Bscores) using Minitab software. The correlation coefficients (CC) between considered descriptors greater than 0.6 are listed in the Appendix A.

The study demonstrated that the binding activity (Bscores) is highly correlated with polarizability (CC = 0.947) and topological diameter (CC = 0.899). Then, by the level of correlation, follow the total surface area (CC = 0.879), non-H-atoms (CC = 0.810), mol. weight (CC = 0.770) and rotatable bonds (0.748). 

The polarizability (*QPpolrz*), topological diameter (*TD*), total surface area, non H-atoms, mol. weight can be used for characterization of pharmacokinetic events, which are connected with such phenomenon as permeation, which depends mainly on size and shape, and on lipophilicity or hydrophobicity, which encodes recognition forces such as hydrophobic interactions, H-bonding capacity, and van der Waals forces (i.e., polarity). They also connected with recognition by proteins that have evolved to be promiscuous (i.e., to bind structurally quite diverse xenobiotics), for example, drug-metabolizing enzymes, xenobiotic transporters, and serum proteins. Here the molecular properties, such as hydrophobicity and H-bonding capacity, play a major role, together with some fuzzy pharmacophores defined by the presence of a small number of recognition groups. 

The descriptor number of rotatable bonds belongs to characteristics of pharmacodynamic events that result from interaction with biological targets, such as receptors, endobiotic-metabolizing enzymes, ion channels, nucleic acids, and so on. Such events are initiated when bioactive agents are recognized by (i.e., bind to) their respective target, a recognition that depends mainly, if not exclusively, on a pharmacophore site within the protein target. In other words, pharmacodynamics events are highly dependent on 3D structure. 

Moreover, the descriptor number of rotatable bonds contains the information about compound’s conformational space. This implicit information is remarkable in suggesting that conformational behavior matters, not only in pharmacodynamics events (drug target recognition), but also from an ADME perspective [52].

#### 3.6.3. The Descriptors Related to Aromaticity

The aromaticity plays a crucial role in drug design. The authors in [56] investigated the impact of aromatic ring count (the number of aromatic and heteroaromatic rings) in molecules against various developability parameters—aqueous solubility, lipophilicity, serum albumin binding, CyP450 inhibition, and human Ether-à-go-go-Related Gene (hERG) inhibition. It was also demonstrated that even within a defined lipophilicity range, increased aromatic ring count leads to decreased aqueous solubility [56].

In the present study, the special focus was made on correlation of aromatic rings and atoms related to FDs. Lipophilicity was expressed as LogP in our study. A weak correlation for aromatic atoms and aromatic rings with LogP (CC = 0.249−0.268) was obtained. The solubility was expressed as LogS and the inverse correlation was obtained for aromatic atoms and aromatic rings with LogS (CC = −0.466/−0.448), which also demonstrated that, in the case of FDs, an increased aromatic ring count leads to decreased aqueous solubility.

Figure 7 represents weight maps for correlated descriptors: aromatic atoms, aromatic rings, sp3-atoms, and stereo centers.

The aromatic atoms correlated with aromatic rings (CC = 0.996). The characterization of the aromatic nature of chemicals here was connected with two main descriptors: sp3-atoms and stereo centers, which are strongly correlated (CC = 0.979). It should be noted that aromatic atoms and aromatic rings have an inverse strong correlation with sp3-atoms and stereo centers (CC = −0.847/−0.852). 

Descriptors: sp3-atoms and stereo centers were used to describe the saturation level of FDs, which are separately discussed in the article below.

#### 3.6.4. The Role of LogP on Properties of FDs Inversely Correlated with Polar Surface Area

The weight maps of cLogP, which is inversely correlated with relative PSA (CC = −0.861) and polar surface area (CC = −0.797), are shown in the Figure 8.

The most commonly used measure of lipophilicity is the LogP parameter. This is, the partition coefficient of a molecule between aqueous and lipophilic phases, usually octanol and water. Lipophilicity is one of the most important physicochemical properties of a drug. It plays a role in solubility, absorption, membrane penetration, plasma protein binding, distribution, CNS penetration and partitioning into other tissues or organs, such as the liver, and has an impact on the routes of clearance. It is important in ligand recognition, not only to the target protein, but also in CYP450 interactions, hERG binding, and PXR-mediated enzyme induction.

The LogP value of a compound, which is the logarithm of its partition coefficient between n-octanol and water log (c_octanol_/c_water_), is a well-established measure of the compound’s hydrophilicity. Low hydrophilicities and; therefore, high LogP values cause a poor absorption or permeation. Both descriptors characterize the *pharmacokinetic* events in a living organism.

It can be concluded that the FDs with a high relative PSA have a low value of LogP. This feature is important in the process of the selection chemicals suitable for drugs.

The authors in [57] reported that increased LogP improves the permeability of chemicals. However, increasing the LogP decreases the solubility and increase the toxicity.

### 3.7. Investigation of Level of Hydrogenation (Saturation) of FDs

Next, the dataset of 169 FDs used in the study was separated into 75 unsaturated FDs and 94 saturated FDs. According to the IUPAC nomenclature [58,59], the fully saturated fullerenes, like C_60_H_60_, belong to fullerenes.

Figure 9 demonstrated that saturated FDs can be separated from unsaturated FDs using the stereo center and sp3-atoms descriptors.

These two descriptors are crucial in drug design. According to Lovering et al. [60], the Fsp3 (carbon bond saturation) is defined as the number of sp3 hybridized carbons/total carbon count. This descriptor correlates with melting point and solubility. The presence of stereo centers as well as a number of chiral centers are descriptors of complexity [60]. Saturation or hydrogenation depends on the amount of hydrogen atoms in FDs, and there is not clear separation on saturated and unsaturated. Only the lower limits are well determined. For saturated FDs, the limit corresponds to sp3 greater than 60. For unsaturated FDs, it corresponds to sp3 = 0. 

Thus, the lowest sp3 value = 0 which corresponds to **FD168** (C60) and **FD50** (C70) with zero sp3 atoms. It should be noted that the pristine fullerene C60 is composed of sp^2^ hybridized carbon nanostructures. Since the discovery of C60 by Kroto et al., the zero-dimensional carbon nanostructure has attracted increasing attention from scientists due to its unique structure and physical properties. A fullerene can act as an electron acceptor because the fullerene molecule requires that the C–C bonds interact through bent sp^2^ hybridized carbon atoms, which leads to a strained structure with good reactivity. The curved π-conjugation of C60 within the unique sp^2^ hybridized carbon atoms shows both π-character and substantial s-character, which is remarkably different from the planar π-conjugation within graphite and planar polycyclic aromatic hydrocarbons that are solely of π-character [61]. Due to the unsaturated character of the C–C bonds in the fullerene, there are plenty of electronic states to accept electrons from appropriate donors forming donor–π–acceptor combinations [62]. In drug design, two simple and interpretable measures of the complexity of molecules, prepared as potential drug candidates, were proposed. The first is carbon bond saturation, as defined by fraction sp3 (Fsp3), where Fsp3 = (number of sp3 hybridized carbons/total carbon count). The second is simply whether a chiral carbon exists in the molecule. It was demonstrated that both complexity (as measured by Fsp3) and the presence of chiral centers correlate with success as compounds transition from discovery, through clinical testing, to drugs. Moreover, it was shown that saturation correlates with solubility, an experimental physical property important to success in the drug discovery setting. 

In the study of [63] it was assumed that compounds with higher solubility, higher permeability, and lower protein binding receive higher developability scores, whereas compounds with lower solubility, lower solubility, and higher protein binding receive lower developability scores. Low developability compounds, in turn, are associated with higher values of the aromatic descriptors and lower values of Fsp3. These descriptors are especially important for the large and lipophilic molecules (MW > 400, cLogP > 4). 

It should be pointed out that FDs molecular weight (MW) is in the range of large molecules (MW = 720−1659). We supposed that FDs should be considered in drug design separately, according to unique properties and structure [64]. The special focus should be directed to sp2 hybridization of carbon in the molecule of fullerene C60.

It was demonstrated above that there is a strong correlation between sp3-atoms and stereo centers (CC = 0.979). Figure 10 demonstrated the inverse correlation between aromatic atoms and sp3-atoms (CC = −0.852), while number of saturated FDs was correlated with sp3-atoms (CC = 0,978) and inversely correlated with aromatic atoms (CC = −0.846). 

In drug design [40,56,65] it is more desirable to shift the balance toward less aromatic and more aliphatic characteristics [56]. It is promisingly to study FDs because their structure contains sp2 carbons. The descriptors of sp3 atoms and inversely correlated aromatic atoms can help in the search for the most promising compounds suitable in drug design. 

The question was aroused about the differences in the binding activity of saturated vs. unsaturated FDs. The saturated and unsaturated FDs having the same substituent (functional group) were selected from our dataset, and were placed in the Appendix A. Appendix A contains the structural formula of considered FD molecules and functional groups, molecular formula, values of descriptors: *QPpolrz*, *TD* and Av_BScores. 

In most cases the saturated fullerenes with the same functional groups had a larger binding affinity than the unsaturated fullerenes. The considered FDs had differences between saturated and non-saturated ones in the range of Bscore 93.99–175.79.

## 4. Conclusions 

The goal of the current study was to characterize the binding activity related to 169 FDs using various cheminformatics tools. It was assumed that the global tendency in binding activity and the application of the special molecular descriptors used in drug design related to topological, electrical, and others properties of studied molecules can contribute to pharmacology or toxicology.

In this study, a classification of FDs was created based on bonding of substituent groups to the C60 core. The following substituent groups were considered:Single bond containing alkyl groups;Cyclopropane three-membered ring;Pyrrolidine (five-membered ring) or pyridine (six-membered) heterocycles containing nitrogen;Six-membered (cyclohexane) ring);Benzene (aromatic six-membered) ring;Fused pair of six-membered rings;Bridged bicycle rings.

Several separate analyses were performed in this study. Thus, a Kohonen map was developed based on 57 of the most important proteins to classify them to certain classes. 

In addition, the analysis of binding activity of each class of FDs and inside classes was done. The overall characteristics demonstrated that the most active FDs have the longest chain of substituents. Benzene, pyridine, and others aromatic rings also contributed to the highest binding activity, as well as the presence of cycle groups. 

The lowest value of binding activity corresponds to pristine fullerene **FD168** (C60). Thus, the fullerene C60 possesses the lowest values of total surface area, molecular weight, rotatable bonds, electronegative atoms, sp3 atoms, polarizability, and topological diameter.

Then the CPANN analysis was done based on 27 descriptors, followed by the consensus model based on 10 descriptors. These models highlighted the importance of the following descriptors. It was shown that polarizability (*QPpolrz*), topological diameter (*TD*), total surface area, non H-atoms, and mol. weight are highly correlated with binding activity. These descriptors were used for characterization of pharmacokinetic events. The number of rotatable bonds descriptor contains information on a compound’s conformational space and is highly correlated with binding activity of FDs. This descriptor characterizes the pharmacodynamic events dependent on interaction with biological targets.

In the study the analysis of 25 drug-like descriptors was made and it was shown how descriptors related to sp3 atoms and stereo centers, inversely correlated with aromatic atoms, helped to separate the hydrogenated (saturated) FDs from unsaturated. This information might be useful for selection of FDs for hydrogen storage technology when at the stage of searching for proper FDs. In addition, the role of LogP was analyzed and how it influences on the binding activity of FDs. 

The results obtained in this study paves the way to study the clusters of proteins responsible for different diseases. Moreover, the investigation of clusters of different proteins enable the prediction of potential FDs as antiviral agents, enzyme inhibitors, ion channel blockers, neuroprotective agents, DNA breakers, photosensitizers in photodynamic therapy, fullerene-specific antibodies and nucleic acid binders, and free radical scavengers. The heatmap of 169 FDs related to 1117 proteins enables the exploration of the possible side effects of FDs in living organisms. 

## Figures and Tables

**Figure 1 nanomaterials-10-00090-f001:**
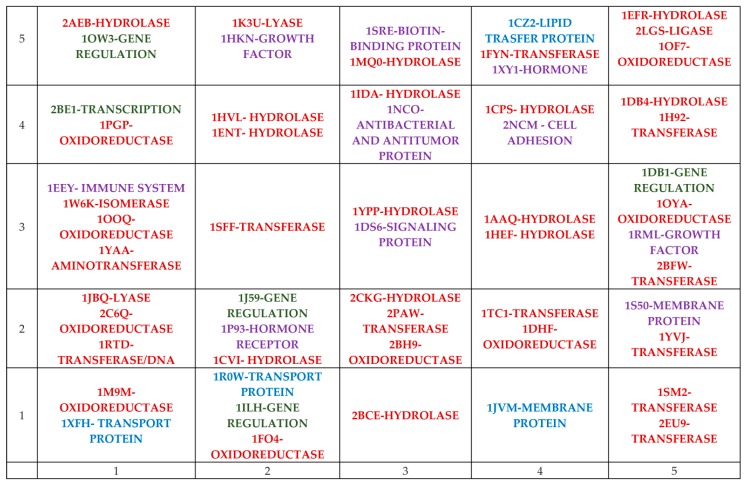
The Kohonen map (5 × 5) with distribution of 57 proteins.

**Figure 2 nanomaterials-10-00090-f002:**
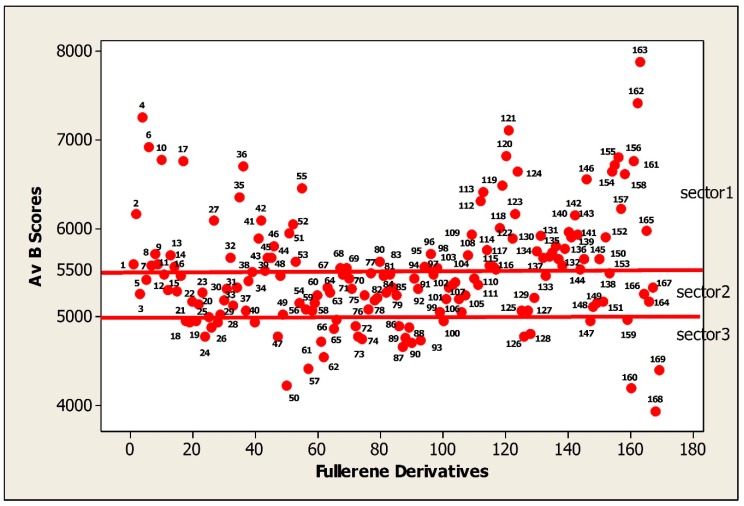
The distribution of fullerene derivatives according to their binding activities.

**Figure 3 nanomaterials-10-00090-f003:**
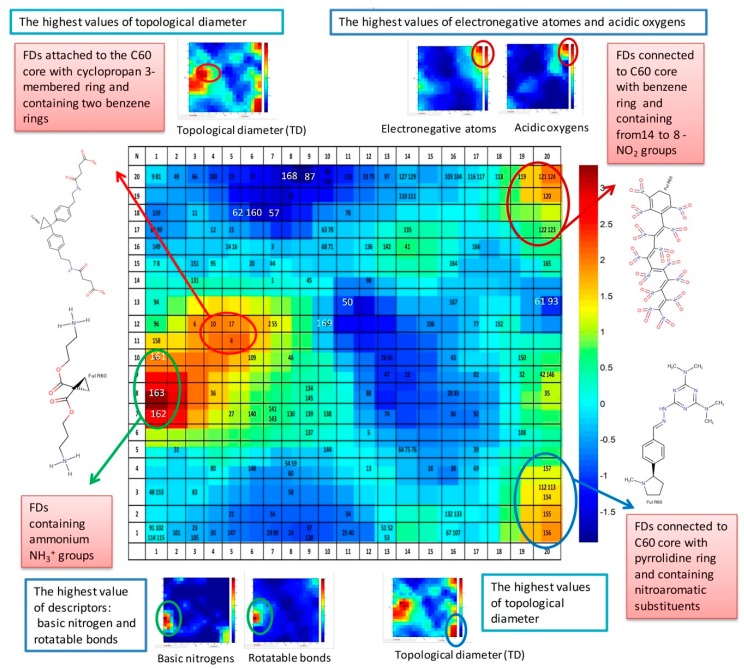
The distribution of FDs in the top map 20 × 20 of the CPANN model based on 27 descriptors overlapped with the output layer with binding activity.

**Figure 4 nanomaterials-10-00090-f004:**
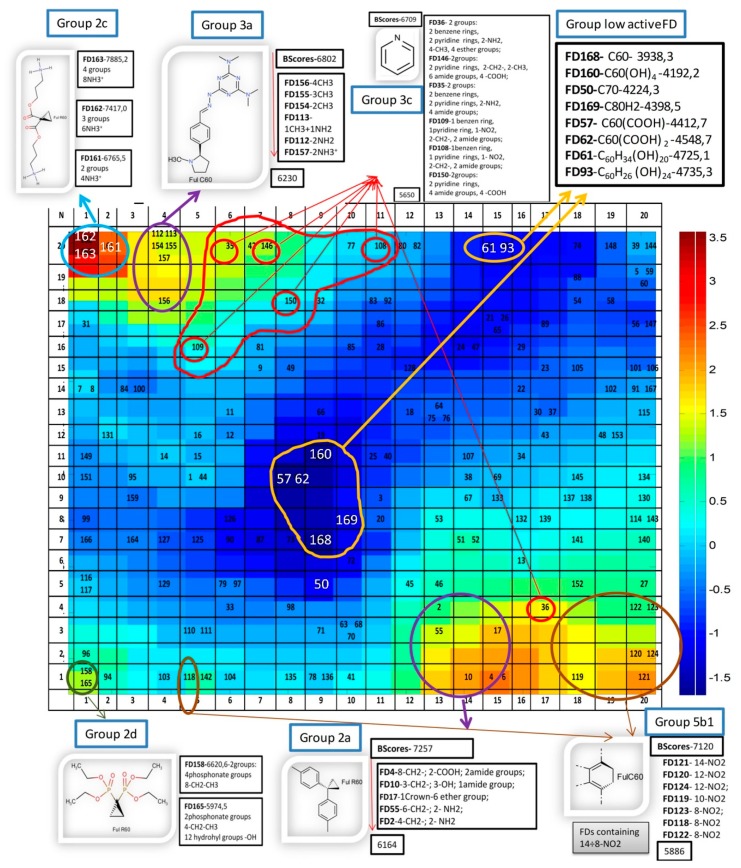
The top map 20 × 20 of the consensus model based on 10 descriptors with distribution of FDs overlapped with the output layer of binding activity (Bscores), with indication of the most active groups of FDs as well as the FDs with lowest activity.

**Figure 5 nanomaterials-10-00090-f005:**
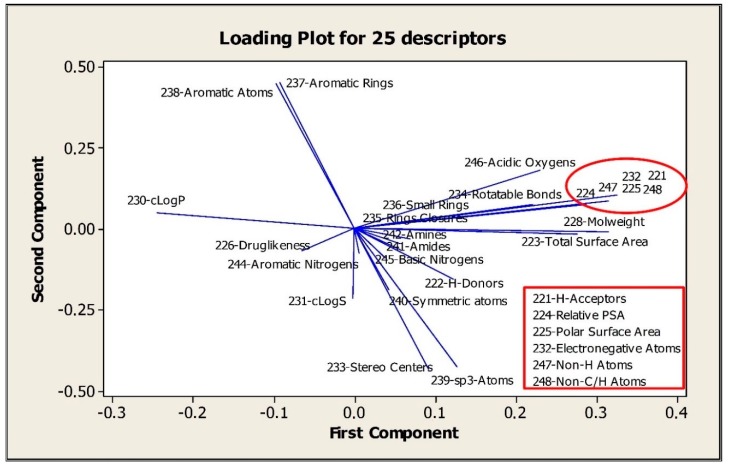
The loading plot for 25 drug-like descriptors.

**Figure 6 nanomaterials-10-00090-f006:**
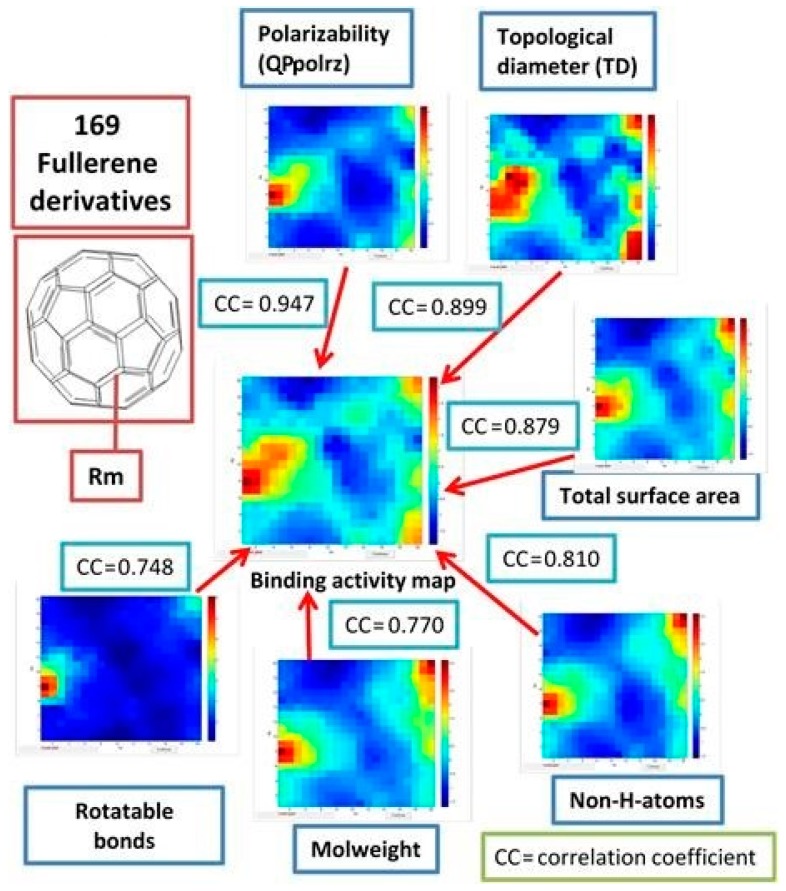
The correlation between drug-like descriptors and binding activity. Weight maps and correlation coefficients (CC).

**Figure 7 nanomaterials-10-00090-f007:**
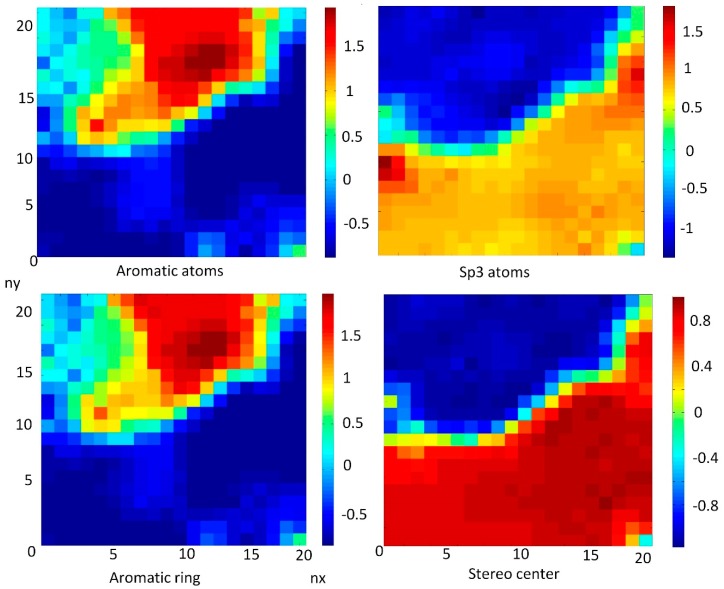
Weight maps for correlated descriptors for aromatic rings and atoms: aromatic atoms, aromatic rings, sp3-atoms, and stereo centers.

**Figure 8 nanomaterials-10-00090-f008:**
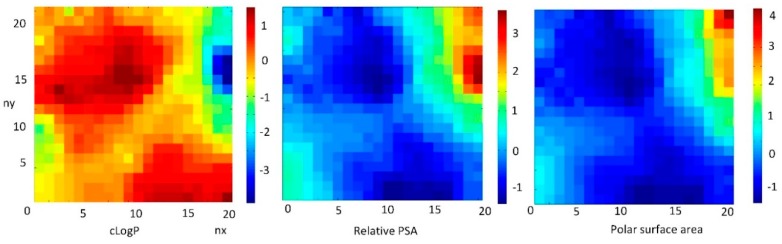
Weight maps of inversely correlated cLogP with polar surface area (PSA) and relative PSA.

**Figure 9 nanomaterials-10-00090-f009:**
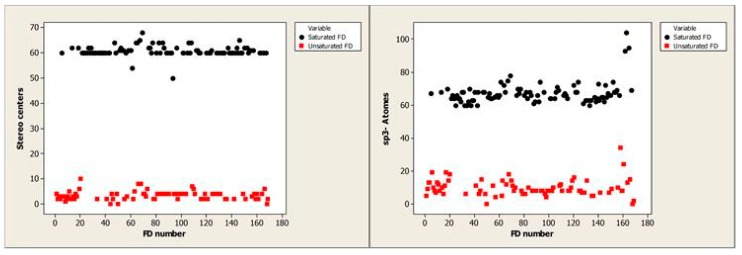
The distribution of saturated and unsaturated FDs depending on the stereo center and sp3-atoms descriptors.

**Figure 10 nanomaterials-10-00090-f010:**
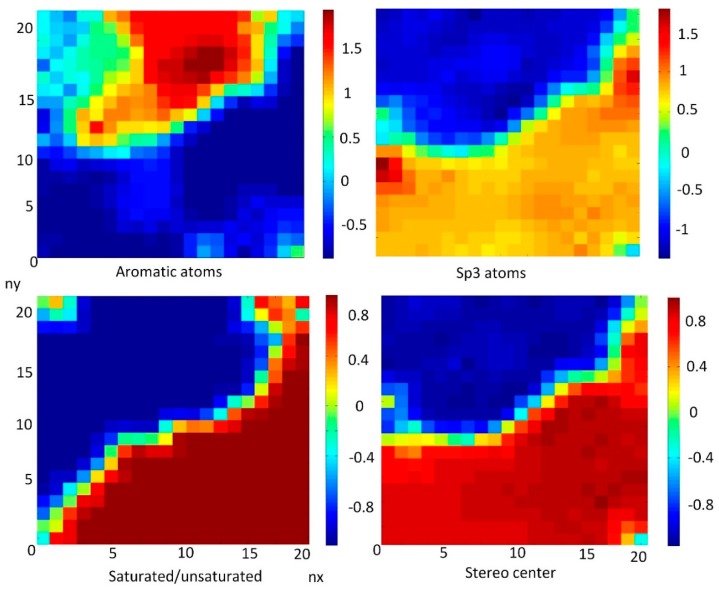
Weight maps of descriptors: aromatic atoms, sp3 atoms, stereo centers with distribution of saturated/unsaturated FDs.

**Table 1 nanomaterials-10-00090-t001:** A short description of groups of proteins investigated in this study related to different functions in organism.

Function of Proteins	Class of Proteins
(a) catalyzing metabolic reactions	*ENZYMES*: OXIDOREDUCTASE; HYDROLASE; ISOMERASES; TRANSFERASE; LYASE; LIGASE
(b) DNA replication	GENE REGULATION; TRANSCRIPTION
(c) responding to stimuli, providing structure to cells and organisms	MEMBRANE RECEPTOR PROTEIN; HORMONE RECEPTOR PROTEIN; IMMUNE SYSTEM PROTEIN; SIGNALING PROTEINS; GROWTH FACTORS; ANTIMICROBIAL ANTITUMOR PROTEINS; CELL ADHESION; BIOTIN-BINDING PROTEIN
(d) transporting molecules from one location to another	TRANSPORT PROTEIN; LIPID TRANSFER PROTEIN

**Table 2 nanomaterials-10-00090-t002:** The correlation between Average sum, Average 110, Average 57, polarizability (*QPpolrz*), and topological diameter (*TD*)

Av_Bscores/Descriptors	Average Sum	Average 110	Average 57	*QPpolrz*	TD
Average sum	1	0.995	0.977	0.947	0.899
Average110	0.995	1	0.974	0.964	0.885
Average57	0.977	0.974	1	0.908	0.889
*QPpolrz*	0.947	0.964	0.908	1	0.837
TD	0.899	0.885	0.889	0.837	1

**Table 3 nanomaterials-10-00090-t003:** The summary table for the least active fullerenes and fullerene derivatives without functional groups or with a minimal number of them.

FD_ID	3D Structure	Mol. Formula and/or Functional Groups	Binding Score Bscores	Difference in Bscores from C60
**FD168**	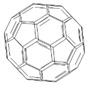	C_60_	3938.3	0
**FD50**	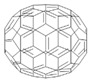	C_70_	4224.3	286
**FD169**	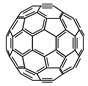	C_80_H_2_	4398.5	461
**FD160**	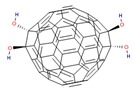	C_60_H_4_O_4_C_60_(OH)_4_4 hydroxyl groups –OH	4192.2	254
**FD57**	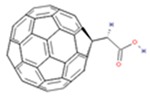	C_62_H_2_O_2_C_60_-CH-COOH	4412.7	475
**FD61**	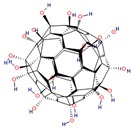	C_60_H_54_O_20_C_60_H_34_(OH)_20_20 hydroxyl groups –OH	4725.1	787
**FD93**	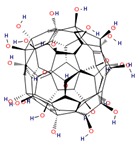	C_60_H_50_O_24_C_60_H_26_(OH)_24_24 hydroxyl groups –OH	4735.3	797

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
