# Peer review of "A Comprehensive Cheminformatics Analysis of Structural Features Affecting the Binding Activity of Fullerene Derivatives"

_nanomaterials, 2020, doi:10.3390/nano10010090_

Round 1

Reviewer 1 Report

This manuscript provided a cheminformatics analysis in fullerene derivatives focusing on their structural features. The CPANN and self-organizing Kohonen network were used in the analyses. This study included 169 fullerene derivatives and 1117 proteins. I found this work novel and topical. Results should be useful for researchers and pharmacologists. However, I would like to give some comments on improving the presentations.

In Figure 1, the 1117 proteins were reduced to 57. It is better to provide a reference (literature) instead of supplementary material regarding the Kohonen map and related algorithm. Figure 2 is very difficult to view because the overlapping of the labels. The authors may consider using a smaller font, bigger figure or a table to present the data. Figure 3: It is difficult to view the details of the figure. The authors can consider using the white color fonts on basically the blue MATLAB background. What is the scale bar representing? Similar issues happened in Figure 4. Resolution of Figure 5 should be increased. Again, sub-figures in Figs. 7, 8 and 10 are too small to view.

Author Response

Thank you for comments for improvement of manuscript.

The answer is in attachment.

Reviewer 2 Report

I believe the ms is suitable for publication before some minor concerns.

I suggest to modify the conclusions section in order to put more emphasis on the obtained results as well as on the possible future applications.

I also suggest to check for the english.

Author Response

Thank you for comments for improvement of manuscript.

The answer is in attachment
